# Research on an Improved Economic Value Estimation Model for Crop Irrigation Water in Arid Areas: From the Perspective of Water-Crop Sustainable Development

**Miaosen Ma [1,2,]*** and **Min Zhao [1,2]**

1    Business School, Hohai University, Nanjing 211100, China; zhaomin3451@sina.com
2    Water Resources and Sustainable Development Research Center of Jiangsu Province, Nanjing 211100, China
*    Correspondence: mamiaosen223@163.com

**Abstract:** This paper aimed to construct an improved economic value estimation model (EVIW model) to assess the economic value of water, which plays an important role in the sustainable development of crop planting and irrigation design, especially in arid areas lacking water resources. Firstly, the current EVIW model was based upon improvements and adjustments to the cost-benefit analysis models of previous researchers. Then, to elaborate the whole process of estimation, an empirical study based on the data of Yanqi Basin was conducted. Subsequently, in order to verify the accuracy of the EVIW model, the economic value of irrigation water in this study area was estimated for a second time using the benefit sharing coefficient method. It was concluded that the estimated results of the current EVIW model are in good agreement with those of the traditional benefit sharing coefficient model. The estimation results of the economic value of irrigation water were found to be highly acceptable in terms of accuracy and scientific rigor.

**Keywords:** EVIW model; water economic value; crop irrigation water; arid area

## 1. Introduction

With the rapid development of society and continuous population growth, the water demands for the irrigation of arid areas is increasing. Therefore, the contradiction between the increasing demand for water resources and the inadequate availability of water resources in arid areas has become more and more obvious, which largely restricts the sustainable development of "economy-society-ecology" in these regions. As the agricultural economy in arid areas is severely constrained by a shortage of water, the maximization of the value of irrigation water is of great significance to agricultural development in arid areas. For example, to achieve the goal of sustainable utilization, the economic value of water can be used as a starting point for the optimal allocation of water for crop irrigation and efficient planning of the crop planting area. In turn, this can provide a basis for improving the utilization efficiency of irrigation water.

The economic value of water resources usually refers to an added value of benefits that can be measured by the available existing currencies and created by per unit of water obtained from natural storage space. This paper aimed to create an effective model to reasonably estimate the economic value of irrigation water by means of mathematical analysis, thereby providing a research basis for the maximization of the economic value of water resources, efficient use of irrigation water, and planning of crop planting. Furthermore, the model created in this paper provides an effective technical basis for "water-crop" sustainable development and regional economic stability in arid areas.

Several researchers have previously employed fundamental economic theories for the economic valuation of water use. Young (1985) illuminated theories on the economics of water resources, such as the definitions of the long-run and short-run value of water and their distinctions. The "equimarginal principle" involved in the maximization of efficiency and utility of water resource development and allocation were also explained in this work. Young also interpreted some theories about how economic benefits relate to location, quantity, and time, and discussed the influence of factors like geographic location on benefits estimates [1]. Ward and Michelsen (2002) outlined and explained a series of issues to elaborate the relevant application of modern economic theories and principles for measuring the economic value of water, in order to guide water management and other related decision-making practices using cost-benefit analysis [2]. They reviewed theories such as the definition and dimensions of water use, the amount of water being valued, targeting values to policies, and physical interactions in the use of water. Bate and Dubourg (1997) performed a net-back analysis on the demand for irrigation water in the East Anglia region of the U.K. The simple net-back model was introduced first, before existing data was used to estimate farmers' willingness to pay for irrigation water, both with and without the effects of agricultural and trade subsidies. This estimate was then compared with estimates of the long-run marginal cost of water [3]. Merrett (2002) provided a clear and succinct account of the link between the cost of irrigation water for farmers and the volume of water that they use, in order to discuss how the use of water for farmers was impacted by the cost of irrigation water [4].

Many studies have been conducted focusing on the economic valuation of water used for different purposes. Frederick et al. (1996) divided the application of fresh water into four kinds of withdrawal and instream uses and estimated the value of water in these applications. In addition, the estimation of value on irrigation water was carried out using the data for 22 crops in different regions of the United States [5]. Annes (2015) introduced the water distribution situation of Colorado River Basin and discussed how to estimate the economic value of water in different regions of the Colorado River Basin. She estimated the value of agricultural, urban, and industrial water uses in different regions of the basin [6]. Schuster et al. (2012) also assessed the value of water in agriculture by making use of the NRTW method and conducted assessments of the practical applications in the Southwestern US and Northwestern Mexico [7]. Hassan-Esfahani et al. (2015) identified the role played by economic valuation in water management, types of water values, and methods for estimating producers' and consumers' water values [8].

The above literature elaborates several methods of estimating the economic value of water from different angles. The most direct method is a comparison of recent market transaction prices for similar types of water transfers in the same area. However, this approach requires the availability of and collection of actual water transaction data in a specific region. Another approach used to assess the value of agricultural water is using water-crop production functions, based on the relationship between irrigation water and crop yield. However, this approach requires assumptions to be made about the level of other crop production inputs, including fertilizers, pesticides, and labor. Furthermore, these methods are labor- and data-intensive and limited to locations and crops where accurate, up-to-date water-crop production functions are available.

To avoid the above shortcomings, the current EVIW model was constructed by improving the NRTW model put forward by Schuster, which is based on cost-benefit analysis and intends to find the net returns of agricultural water use brought to the crops. However, this method only deals with the whole economic value of agricultural water use, ignoring the differences between different kinds of crops. Besides, the overall cost is used for calculation without concrete decomposition. The EVIW model includes a series of improvements to make the accounting of each income and cost item more accurate and practical. For example, as the corresponding cost items for different kinds of crops are quite different, it was necessary to keep useful cost items and remove unnecessary ones. In other words, it is an improved approach combining cost-benefit analysis with the cost decomposition method for assessing the economic value of irrigation water for different crops. It estimated the on-farm economic

value of water in crop production by subtracting variable production costs (excluding water costs) from gross revenues per acre. In other words, the value of water was estimated as the difference between gross crop revenues and non-water input costs in detail, i.e., the economic value of a unit of irrigation water for each crop can be estimated by this new model. The benefits of the EVIW model are: (1) it is accessible across broader scales and less expensive than the crop-water production function method; (2) it can be applied to the many regions where data is not available for the market price comparison method.

Subsequently, in order to verify the accuracy of the EVIW model, the economic value of irrigation water (EVIW) in the study area was estimated once again by using the benefit sharing coefficient method and the results were compared with those of the EVIW model. The benefit sharing coefficient method (BSCM) refers to the ratio of crop benefit increase caused by irrigation to total crop yield benefit increase. It is an important parameter for evaluating the economic benefits and making investment decisions in irrigation projects [9].

## 2. Construction of the EVIW Model

The core aim of this study was to construct a new, comprehensive, and appropriate model, named the EVIW model, based on the theories of previous models, in order to estimate the economic value of crop irrigation water in arid areas. Then an empirical study using the data of Yanqi Basin was conducted to elaborate the whole process of estimation. Figure 1 shows the mechanism flow chart of the EVIW model. As shown in Figure 1, the EVIW model was developed according to following five modules: The calculation of gross revenues resulting from irrigation; the calculation of variable costs for irrigation areas; the calculation of net revenues resulting from irrigation; the calculation of total gross irrigated water requirement; and the estimation of the economic value of irrigated water.

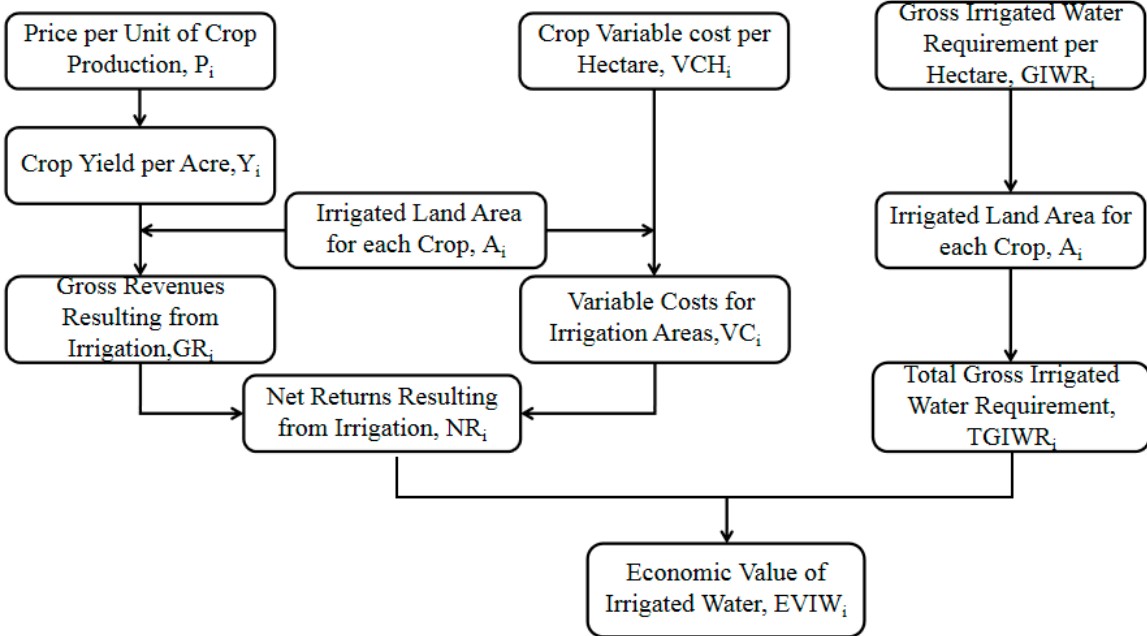

**Figure 1.** Mechanism flow chart of economic value of irrigation water (EVIW) model.

### 2.1. Gross Irrigated Water Requirement

The gross irrigated water requirement (GIWR) is the depth of irrigation water required, when irrigated land, irrigation efficiency, precipitation, crop evapotranspiration, and other related uses, such as water required for leaching and frost protection [10], are considered. In the present study, GIWR was estimated using a deterministic GIWR model that uses mathematical representation of the physical process to estimate the state of the system during a given time period. The GIWR model keeps

track of soil moisture by accounting for daily soil water inputs, such as precipitation and irrigation, and daily soil water outflows from evapotranspiration.

A monthly GIWR model was set up by following guidelines from the USDA Soil Conservation Services [10] and University of Arizona Department of Agriculture and Biosystems Engineering to estimate GIWR for all crops in each groundwater sub-basin.

The GIWR model has two steps. Firstly, it computes the reference crop evapotranspiration ($ET_0$) using the Penman-Monteith equation with the meteorological data of the research area [11]. To obtain crop evapotranspiration for crop $i$ ($ET_i$), crop coefficient ($K_c$) which is a function of crop type and stage of growth was used. $K_c$ for the major crops were directly obtained from USDA Agriculture Soil Conservation Service and modified for each growing stage based on typical planting and harvesting dates for the crops [12]. Then, $ET_i$ can be calculated as:

$$ET_{Ci} = K_{Ci} \times ET_0 \tag{1}$$

GIWR for crop i ($GIWR_i$) can be determined by:

$$GIWR_i = \frac{ETc_i - EP + W_i}{e} \tag{2}$$

where, $EP$ is effective precipitation, which is obtained by multiplying rainfall $P$ with rainfall coefficient $a$. Rainfall $P$ can be obtained by yearly statistical data. For rainfall coefficient $a$, when $P \leq 5$ mm, $a = 0$; when $P = 6$–$50$ mm, $a = 0.8$–$1.0$; when $P > 50$ mm, $a = 0.70$–$0.80$ [13]. $W_i$ is other water uses; $e$ is the irrigation efficiency. Total water volume for irrigation for crop $i$ in cropland then can be computed by multiplying its irrigated area. For this study, the leaching requirements for salinity control were assumed to be 10% of $ET_i$. Furthermore, 75% of irrigation efficiency was used as it is typical value in this research [14].

### 2.2. Gross Revenues

Gross revenues represent the value of total output of crops irrigated by each groundwater sub-basin. The calculation formula for gross revenues is expressed as:

$$GR_i = P_i \times Y_i \times A_i \tag{3}$$

where $i$ is crop type; $GR_i$ is gross revenues for crop $i$; $P_i$ is price per unit for crop $i$; $Y_i$ is yield per hectare for crop $i$; $A_i$ is the irrigated land area for crop $i$.

### 2.3. Variable Costs

The variable costs of crop production consist of labor costs, chemical costs, machinery costs, pre-harvest costs, custom and pick up use. The variable costs can be calculated as follows:

$$VC_i = VCH_i \times A_i \tag{4}$$

where $i$ is crop type; $VC_i$ is variable cost for crop $i$; $VCH_i$ is variable cost per hectare for crop $i$; $A_i$ is the irrigation land area for crop $i$.

### 2.4. Net Returns

The net returns ($NR_i$) of crop $i$ in irrigated areas is determined by gross revenues and corresponding variable costs as shown in the following equation.

$$NR_i = GR_i - VC_i \tag{5}$$

where $i$ is crop type; $NR_i$ is net revenues for crop $i$; $GR_i$ is gross revenues for crop $i$; $VC_i$ is variable cost for crop $i$.

### 2.5. Economic Value of Irrigation Water

The economic value of irrigation water for crop $i$ is determined by the total net return of crops and the value of irrigation water use in irrigated areas, as shown in the following equation.

$$EVIW_{E,i} = \frac{NR_i}{TGIWR_i} = \frac{NR_i}{GIWR_i \times A_i} \tag{6}$$

where $i$ is crop type; $EVIW_{E,i}$ is the economic value of irrigation water for crop $i$ by EVIW model; $NR_i$ is the net returns for crop $i$; $TGIWR_i$ is the total gross irrigated water requirement for crop $i$; $GIWR_i$ is the unit gross irrigated water requirement for crop $i$; $A_i$ is the irrigated land area for crop $i$.

## 3. Control Group Model-Benefit Sharing Coefficient Method

Because of its high practicability and standardization, the benefit sharing coefficient method has been widely used for estimating the economic value of water resources since last century [15]. The common agricultural irrigation benefit sharing coefficient ($\varepsilon_{Ir}$) is calculated by the test parameter method, while the water supply benefit allocation coefficient for the other water use activities is calculated by the cost proportion method. The benefit sharing coefficient method is highly feasible, but it needs to be determined reasonably. It is carried out in two steps: firstly, calculating the benefit of crop irrigation allocation; secondly, calculating the economic value of crop net irrigation water [16].

The economic value of irrigation water for crops is calculated by using benefit sharing coefficient model. The formulas are as follows:

$$EVW_{B,i} = \frac{TVW_{Ir,i}}{Q_i} = \frac{TVW_{Ir,i}}{TGIWR_i} = \frac{TVW_{Ir,i}}{GIWR_i \times A_i} \tag{7}$$

where $i$ is crop type; $EVW_{B,i}$ is the economic value of irrigation water for crop $i$ by BSCM method; $TVW_{Ir,i}$ is the sharing benefits of irrigated water for crop $i$; $Q_i$ is the total volume of irrigation water for crop $i$; $TGIWR_i$ is the total gross irrigated water requirement for crop $i$; $GIWR_i$ is the unit gross irrigated water requirement for crop $i$; $A_i$ is the irrigated land area for crop $i$.

Meanwhile, the formula for calculating the sharing benefits of irrigated water for crops is as follows:

$$TVW_{Ir,i} = \varepsilon_{Ir,i} \times P_i \times Y_i \times A_i \tag{8}$$

where $i$ is crop type; $\varepsilon_{Ir}$ is the irrigation benefit sharing coefficient for crop $i$; $P_i$ is price per unit for crop $i$; $Y_i$ is yield per hectare for crop $i$; $A_i$ is the irrigated land area for crop $i$.

In addition, the irrigation benefit sharing coefficient is determined by the field comparative experiment method. The experimental areas with uniform soil and hydro-geological conditions are selected and divided into several plots for comparative experiments [17]. The treatment was as follows:

Plot No.1: general level agricultural measures without irrigation, the yield is named as $Y_1$;

Plot No.2: general level agricultural measures with irrigation, the yield is named as $Y_2$;

Plot No.3: high level agricultural measures without irrigation, the yield is named as $Y_3$;

Plot No.4: high level agricultural measures with irrigation, the yield is named as $Y_4$;

Based on the experimental results $Y_1$, $Y_2$, $Y_3$, $Y_4$, the irrigation benefit sharing coefficient $\varepsilon_{Ir,i}$ can be obtained by Equation (9).

$$\varepsilon_{Ir,i} = \frac{(Y_2 - Y_1) + (Y_4 - Y_3)}{2(Y_4 - Y_1)} \tag{9}$$

## 4. Case Study

### 4.1. Overview of Study Area

The representative Yanqi Basin was selected as the study area in this research. Yanqi Basin (41°42′35″–42°20′05″ N, 85°56′33″–87°29′50″ E) is located in the north of Xinjiang Bayingol Mongolian Autonomous Prefecture, and the middle of the Tianshan Mountains and Kuruktag Mountains. The overall area is about 6865 km². It includes Hejing County, Heshuo County, Yanqi County and Bohu County (also known as the North Four Counties), Xinjiang Construction Corps 21st regiment, Xinjiang Construction Corps 27th regiment and Xinjiang Construction Corps 223rd regiment. According to the statistical data of "Yanqi Yearbook", the main crops in Yanqi Basin are wheat, beet, corn, forage crop, oil-bearing crop, cotton, tomato, fruit crop, rice, hops, and other crops.

### 4.2. Data Sources

The unit price and unit output data of the main crops ($P_i$ and $Y_i$) in Yanqi Basin from 2013 to 2017 were obtained by consulting the "Bayingol Statistical Yearbook 2013–2017". The unit price data for several crops was missing, in these cases we used the data from "Xinjiang Statistical Yearbook 2013–2017" as an alternative.

The irrigated land area data of main crops ($A_i$) in Yanqi Basin from 2013 to 2017 were collected from "China Agricultural Statistical Data 2013–2017", "China Water Resources Database Collection" and the documentation of Yanqi agricultural departments. For the missing data, the average values of recent years were used as alternatives.

This paper used the Producer Price Payment Index (PPPI) to estimate the unit variable cost data of the main crops from 2013 to 2017 based on the historical data of 2003. This method has been widely used in crop production cost accounting in the United States. Researchers from the University of California, estimated crop variable cost data in data-missing years by referring to historical data in crop budgets and achieved good results [18,19]. The formula for calculating variable costs per hectare with PPPI is as follows.

$$VCH_{it} = VCH_{it} \times \frac{PPPI_t}{PPPI_{2003}} \tag{10}$$

where $i$ is crop type; $t$ is time; $VCH_{it}$ is the variable cost per hectare for crop $i$ in year $t$; $VCH_{i2003}$ is the variable cost per hectare for crop $i$ in year 2003; $PPPI_t$ is Producer Price Payment Index in Year $t$; $PPPI_{2003}$ is Producer Price Payment Index in Year 2003.

The irrigation benefit sharing coefficient is determined by the field comparative experiment method. After visiting the Yanqi County Water Conservancy Bureau and interviewing local farmers, this study summarized the crop yields per hectare of eight main crops in four scenarios.

### 4.3. Results and Analysis

The comparison line charts of irrigation water economic value estimation for the eight main crops based on two methods are shown in Figure 2.

As shown in Figure 2, on the whole, the EVIW values for the eight main crops based on the EVIW model were close to those of the traditional irrigation benefit sharing method and the errors between the two results for corresponding years are quite small. Only the estimation results of corn and cotton by these two estimation models (methods) had slightly more significant errors, with the highest values of 10.0% and 18.7% respectively in 2013. The error intervals for other crops were generally less than 5%. Except for wheat, the EVIW variation trends of other seven crops by using the two estimation models were very close to each other. The degree of coincidence is very high. Because of the specific value of the irrigation allocation coefficient, the estimation variation range of irrigation benefit sharing method is smaller than that of EVIW method. As in this study the field experiment method was used to estimate the irrigation benefit sharing coefficients and the average values of the experimental observations were used as the unified values of the case interval, the estimation

results were relatively more stable for the irrigation benefit sharing method than the EVIW method. Regarding the latter method, the estimation results for each year were more independent. In addition, the variation ranges of estimation results between adjacent years for wheat, corn and cotton were relatively larger than those for other crops. As the average irrigation water price in Yanqi basin from 2013–2017 was 0.085 RMB/m$^3$, it can be seen that the economic value of all crops exceeded the average water cost.

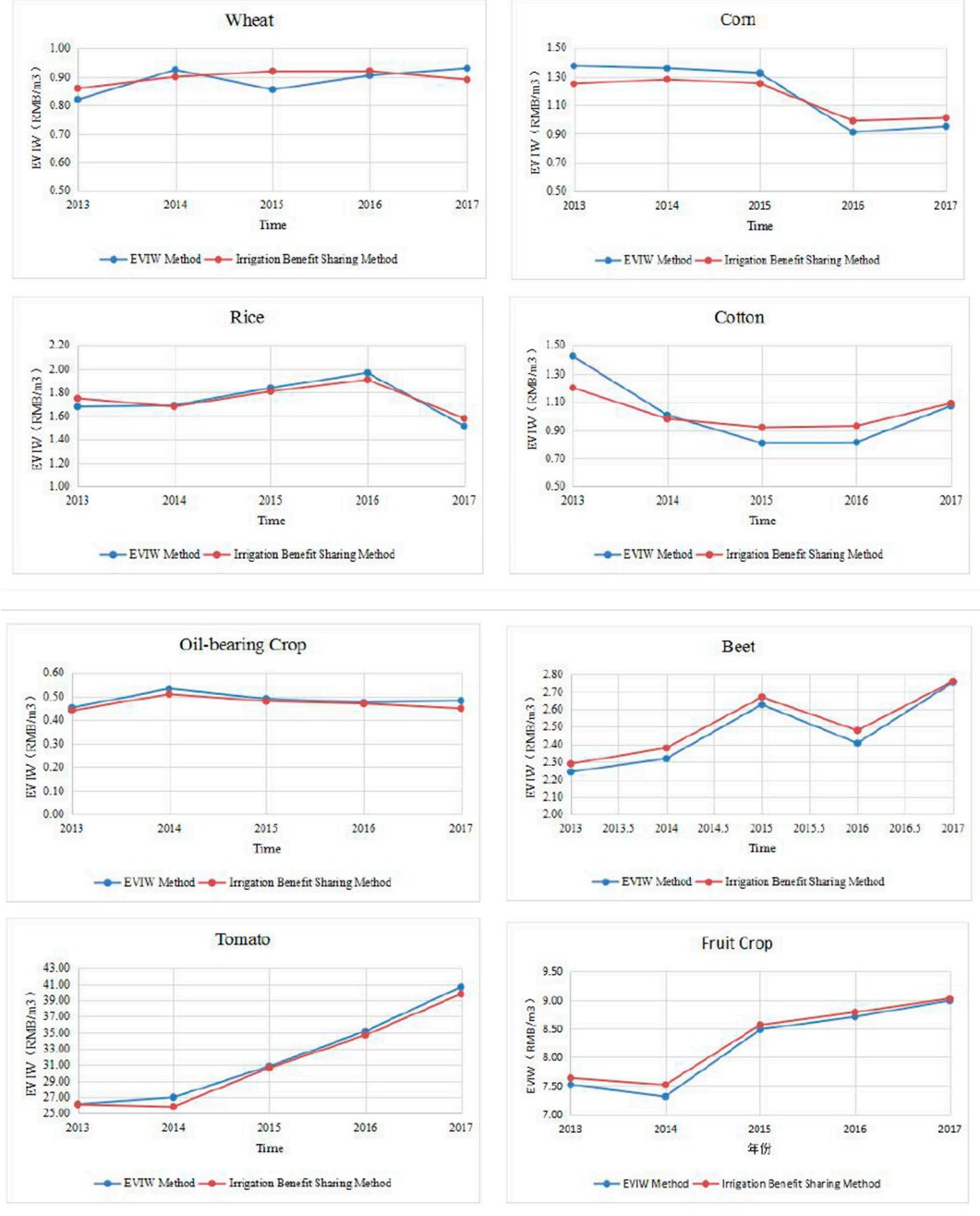

**Figure 2.** Comparative charts of water economic value for eight crops between EVIW method and benefit sharing coefficient method.

## 5. Conclusions

This case study estimated EVIW values in an arid area, the Yanqi Basin, showing that the EVIW model is not only suitable for estimation research with large-scale data, but also suitable for use when

data is partially absent. On the basis of previous research, this paper improved the existing models and constructed a new EVIW model to estimate the economic value of crop irrigation water for arid areas. Subsequently, the economic value of crop irrigation water was estimated again by using the benefit sharing coefficient method as a control. Finally, the estimation results of the EVIW model were compared with those of the traditional irrigation benefit sharing method. Overall, the research allowed several main conclusions, as follows:

The EVIW model makes a contribution to achieving "water-crop" sustainable development and regional economic stability in arid areas, in the following three ways: Firstly, by comparing the economic value of irrigation water of different crops, it allows farmers to choose to plant crops associated with a greater economic value of water, in order to overcome the problem of water shortages and realize the sustainable utilization of farmland. Secondly, when combined with plant planning, the estimated economic value of irrigation water can be used as a judgement criterion for the optimal allocation of crop irrigation water. Thirdly, the estimated economic value of irrigation water can provide a basis for improving the utilization efficiency of irrigation water.

The EVIW model is a combination of the cost-benefit analysis method and the cost decomposition method. A series of improvements were made to make the accounting of each income and cost item more accurate and practical.

The EVIW model considers the gross revenues of crops based on market price and a series of costs generated over crop growth (excluding fixed costs that do not vary with planting area, including environmental costs and social costs). The model uses the economic return on irrigation water brought to crop production as the ultimate criterion for judging irrigation allocation, rationally allocates irrigation farmland and irrigation water, and then reduces the water consumption that produces the minimum economic value to a sustainable utilization level. It is not only a theoretical innovation, but also has high practicability.

The estimated results of the EVIW model are in good agreement with those of the traditional benefit sharing coefficient model. In order to help realize the sustainable utilization of water for the eight main crops in Yanqi Basin, the estimation results of the economic value of irrigation water were highly acceptable in terms of accuracy and scientific rigor However, a few words about the main limitations of the paper are necessary. Specifically, the current approach ignored the water scarcity rents of irrigated crops that should be included in the costs. The approach for water scarcity rents of each crop needs a deep exploration in the future.

**Author Contributions:** Conceptualization, M.M. and M.Z.; methodology, M.M.; software, M.M.; validation, M.M.; formal analysis, M.M.; resources, M.Z.; data curation, M.M.; writing—original draft preparation, M.M.; writing—review and editing, M.M.; visualization, M.Z.; supervision, M.Z.; project administration, M.Z.

**Funding:** This research received no external funding.

**Acknowledgments:** This study was supported by Chinese Scholarship Council "Joint PhD. Student Program" (Grant: 201606710037).

**Conflicts of Interest:** The authors declare no conflict of interest. The funders had no role in the design of the study; in the collection, analyses, or interpretation of data; in the writing of the manuscript, or in the decision to publish the results.

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
