# Peer review of "Research on an Improved Economic Value Estimation Model for Crop Irrigation Water in Arid Areas: From the Perspective of Water-Crop Sustainable Development"

_sustainability, doi:10.3390/su11041207_

Round 1
Reviewer 1 Report
The article submitted suffers from all the main shortcomings of a standard neoclassical analysis. First, it does not at all mention environmental aspects and negative externalities from related agricultural production. Second, the approaches to value setting based on marginal utility of production factors show only seemingly “fair” prices for irrigation water, as they come only from static information and small incremental magnitudes and do not answer questions about broader social fairness that should be solved in order to achieve the real sustainability of agricultural production.
Author Response
Response to Reviewer 1 Comments
Dear professor, thank you very much for your review of the paper. The following are my explanations to your questions.
Point 1: It does not at all mention environmental aspects and negative externalities from related agricultural production.
Response 1: Thank you for your review, dear professor. You got right down to the heart of the matter. First, in regards to this question, I think there is some confusion of the paper title. “The sustainable utilization of” should be removed. This paper is mainly to construct an improved economic value estimation model (EVIW model) for assessing the economic value of water used for crop irrigation, especially in acid areas lacking water resources, so it emphasizes more on how to save water and make the irrigation water values optimized. I.e., the paper is a topic of optimal allocation. Thus, I suggests on the basis of your instructions that the paper title should be adjusted a bit. Second, in EVIW approach, environmental costs, social costs etc. are considered in the cost accounting for EVIW estimation. The model constructed in this paper is an easier way to evaluate the economic value of crop irrigation water, which can give some suggestions for future researches in this field.
Point 2: The approaches to value setting based on marginal utility of production factors show only seemingly “fair” prices for irrigation water, as they come only from static information and small incremental magnitudes and do not answer questions about broader social fairness that should be solved in order to achieve the real sustainability of agricultural production.
Response 2: First, Thank you very much for pointing out this question, your doubt on my approach is really remarkable. After more detailed consideration, I think the EVIW model created in this paper should be an improved model based on Net Returns to Water (NRTW) method and “Cost-benefit” Analysis that commonly used for estimating the economic values of crop irrigation water, and it has its own advantages: ① Compared with NRTW model (same principle with “cost-benefit” analysis), the EVIW model subtracts non-water input costs from gross revenues to estimate the value of irrigated water. Specifically, the irrigation water costs and some fixed costs not varying with the volume of irrigation water are removed from the overall cost. ② The GIWR model is integrated into the EVIW model for the estimation of crop irrigated water requirement, which is useful for calculating the water economic value of different crops. The monthly GIWR model is set up by following guidelines from the USDA Soil Conservation Services and University of Arizona Department of Agriculture and Bio-systems Engineering to estimate GIWR for different crops in each groundwater sub-basins. It takes other water uses of different crops into consideration to make the water requirement calculation for all crops more accurate.
Second, In order to explain why the EVIW model can be a useful method for economic value of irrigation water, the following paragraph is added into second paragraph: The EVIW model is constructed by improving the NRTW model put forward by Schuster. The NRTW model is based on “Cost-benefit” Analysis and intends to find the net returns of agricultural water use brought to the crops. However, this method only deals with the whole economic value of agricultural water use, ignoring the differences between different kinds of crops. Besides, the overall cost is used for calculation without concrete decomposition. The EVIW model is a has taken a series of improvements to make the accounting of each income and cost item more accurate and practical. For example, as the corresponding cost items for different kinds of crops are quite different, it is needed to keep useful cost items and remove unnecessary ones. In other words, it’s an improved approach combining Cost-benefit Analysis with Cost Decomposition Method for assessing the economic value of irrigation water for different crops.
Reviewer 2 Report
This paper uses the Economic Value of Irrigation Water model (EVIW) to estimate the economic value of water and compare it to results from the Benefit Sharing Coefficient Model. The EVIW model subtracts non-water input costs from gross revenues to estimate the value of irrigated water.
The contribution of the paper is presented as showing that the EVIW model produces comparable results to an established method, but is simpler to calculate. The authors do this by presenting graphs of results for the two methods. These graphs indicate that the results are fairly similar.
Although not familiar with either approach, the results seem to support the author's claim that the the EVIW approach produces comparable results. If it is indeed easier to calculate, then I believe this is a publishable contribution to the literature.
This manuscript does need to be edited for grammar and some absent references. For example, on lines 165-167 the authors reference research from UC Davis without a citation. There are also numerous instances of omitted words that decrease the quality of the writing. For example, a sentence on line 66 starts out "GIWR model" instead of "The GIWR model". A careful edit of the paper would increase the clarity.
Author Response
Response to Reviewer 2 Comments
Thank you very much for your review of the paper. According to your suggestions, I’ve made detailed revisions after careful consideration as follows.
Point 1: This manuscript does need to be edited for grammar and some absent references. For example, on lines 165-167 the authors reference research from UC Davis without a citation. There are also numerous instances of omitted words that decrease the quality of the writing. For example, a sentence on line 66 starts out "GIWR model" instead of "The GIWR model". A careful edit of the paper would increase the clarity.
Response 1: ① In regards to the grammar, I asked an English-proficient professor to help me go through the article and mark the wrong parts, and then I corrected all the errors. The modification of language and grammar is highlighted in the revised draft. If necessary, I will ask expert editors from MPDI for language editing. ② For reference absence, I rechecked the manuscript and made some corrections afterwards. For example, the omitted citation [18] and [19] of “reference research from UC Davis” is supplemented both on lines 165-167 and in the reference section. ③ For omitted words that decrease the quality of the writing, I made corresponding corrections during the process of grammar revising.
Reviewer 3 Report
I read with great interest the manuscript entitled “Research on Economic Value Estimation Model for the Sustainable Utilization of Crop Irrigation Water in Arid Areas. The paper is not well organized, the exposition can be improved and the discussion does not do justice to the analysis.
Some of my reservations are, as they appear:
1) General comment, the manuscript will be improved by a professional editing, there are so many language blemishes.
2) Reference style should be either author-date or numbered, not both.
3) Line 7, What you term as EVIW is not new method. It is the residual method (see Young, R. (2005). Determining the Economic Value of Water: Concepts and Methods. Washington, RFF Press.) or net-back analysis (see Bate, R. and W. Dubourg (1997). "A Net-Back Analysis of irrigation Water Demand in East Anglia." Journal of Environmental Management 49: 311-322.)
4) Reference [2)] is missing.
5) Line 30, you can hardly call theories the 12 propositions put forward by Merrett.
6) Line 36, How this paper improves the research models?
7) Line 47, the Benefit Sharing Coefficient Method (BSCM) is not defined.
8) Lines 59, Is it necessary a figure to address the residual method?
9) Lines 108 & 123 What is the difference between equations (6) and (7)?
10) References [10] and [11] do not mention the Benefit Sharing Coefficient Method.
11) Line 143, equation (9) needs to elaboration. How was obtained?
12) Line 163, is the PPPI specific to agricultural product?
13) Line 213, since you are considering a broad interpretation of costs as to include environmental costs you should not ignore the scarcity cost of water. See Kampas, A. and S. Rozakis (2017). "On the Scarcity Value of Irrigation Water: Juxtaposing Two Market Estimating Approaches." Water Resources Management 31(4): 1257-1269.
14) Lines 225- 249, Check the accuracy of the reference’s list.
Author Response
Response to Reviewer 3 Comments
Thank you very much for your review of the paper. According to your suggestions, I’ve made detailed revisions after careful consideration as follows.
Point 1: General comment, the manuscript will be improved by a professional editing, there are so many language blemishes.
Response 1: In regards to the language blemishes and grammar errors, I asked an English-proficient professor to help me go through the article and mark the wrong parts, and then I corrected all the errors. The modification of language and grammar is highlighted in the revised draft. If necessary, I will ask expert editors from MPDI for language editing.
Point 2: Reference style should be either author-date or numbered, not both.
Response 2: According to your instructions, all references in the article are numbered in sequence. On the basis of the first manuscript, several changes are made as follows: ① “USDA Soil Conservation Service 1993” on lines 64 and 70 are replaced by citation number “[10]”. ② “Brown 2005” on line 73 is removed. ③ “USDA-NASS 2010” on line 77 is replaced by citation number “[12]”. ④ “Fox et al. 1993; Annes 2015” is removed, only leaving citation numbers “[14]” there.
Point 3: Line 7, What you term as EVIW is not new method. It is the residual method (see Young, R. (2005). Determining the Economic Value of Water: Concepts and Methods. Washington, RFF Press.) or net-back analysis (see Bate, R. N. and Dubourg, W. R. (1997). "A Net-Back Analysis of irrigation Water Demand in East Anglia." Journal of Environmental Management 49: 311-322.)
Response 3: First, thanks for pointing out this mistake. After detailed reading of these two literature you provide, I think the EVIW model created in this paper is not a new model but an improved model based on Net Returns to Water (NRTW) method and “cost-benefit” analysis that used for estimating the marginal cost of irrigation water: ① Compared with NRTW model (same principle with “Cost-benefit” Analysis), the EVIW model subtracts non-water input costs from gross revenues to estimate the value of irrigated water. Specifically, the irrigation water costs and some fixed costs not varying with the volume of irrigation water are removed from the overall cost. ② The GIWR model is integrated into the EVIW model for the estimation of crop irrigated water requirement, which is useful for calculating the water economic value of different crops. The monthly GIWR model is set up by following guidelines from the USDA Soil Conservation Services and University of Arizona Department of Agriculture and Bio-systems Engineering to estimate GIWR for different crops in each groundwater sub-basins. It takes other water uses of different crops into consideration to make the water requirement calculation for all crops more accurate.
Based on your valuable suggestions, I’ve made the following modifications:① Changing the title to Research on an Improved Economic Value Estimation Model for the Sustainable Utilization of Crop Irrigation Water in Arid Areas; ② making corresponding adjustment of EVIW model interpretation in “Abstract” and main body of the article.
Point 4: Reference [2] is missing.
Response 4: The reference [2] is supplemented in the reference section.
Point 5: Line 30, you can hardly call theories the 12 propositions put forward by Merrett.
Response 5: The description is changed to “Merrett (2002) provided a clear and succinct account of the linkage between the cost of irrigation water to farmers and the volume of water that they use in order to discuss how the use of water for farmers was impacted by irrigation water cost”. “Merrett (2002) introduced 12 theories on the cost of irrigation water ” is removed.
Point 6: Line 36, How this paper improves the research models?
Response 6: In order to explain this problem, the following paragraph is added into second paragraph: The EVIW model is constructed by improving the NRTW model put forward by Schuster. The NRTW model is based on “Cost-benefit” Analysis and intends to find the net returns of agricultural water use brought to the crops. However, this method only deals with the whole economic value of agricultural water use, ignoring the differences between different kinds of crops. Besides, the overall cost is used for calculation without concrete decomposition. The EVIW model is a has taken a series of improvements to make the accounting of each income and cost item more accurate and practical. For example, as the corresponding cost items for different kinds of crops are quite different, it is needed to keep useful cost items and remove unnecessary ones. In other words, it’s an improved approach combining Cost-benefit Analysis with Cost Decomposition Method for assessing the economic value of irrigation water for different crops.
Point 7: Line 47, the Benefit Sharing Coefficient Method (BSCM) is not defined.
Response 7: The Benefit Sharing Coefficient Method (BSCM) refers to the ratio of crop benefit increase caused by irrigation to total crop yield benefit increase. It is an important parameter for evaluating the economic benefits and making investment decisions of irrigation projects. The definition is added on line 61 of the revised manuscript.
Point 8: Lines 59, Is it necessary a figure to address the residual method?
Response 8: The figure represents the flow chart of overall EVIW model which has the same principles of residual method. In order to distinguish the differences between these two, the improvement made for EVIW model is re-elaborated in the second paragraph.
Point 9: Lines 108 & 123 What is the difference between equations (6) and (7)?
Response 9: Both equation (6) and (7) are used for calculating the economic value of irrigation water. Equation (6) is for EVIW model, while equation (7) is for BSCM method. In order to distinguish them, I changed the EVIW values of the two models into EVIWE,i and EVIWB,i respectively.
Point 10: References [10] and [11] do not mention the Benefit Sharing Coefficient Method.
Response 10: The references [10] and [11] are wrongly cited. The right ones are as follows:
[10] Shouhua C., and Zhanyu Z.(2008). Research Situation on Irrigation Benefit Sharing Coefficient and New Calculation Methods. Water Saving Irrigation, 2008(2), 25-27.
[11] Fanghua H., and Jianyong Z.(2001). Economic Benefit Analysis of Irrigation Project in Beijing. Water Resources Protection, 2001(2), 4-6.
Because the reference sequence is changed, references [10] and [11] are changed into [15] and [16] respectively in the revised manuscript.
Point 11: Line 143, equation (9) needs elaboration. How was obtained?
Response 11: Equation (9) uses Producer Price Payment Index (PPPI) to estimate the unit variable cost data of main crops from 2013 to 2017 based on the historical data of 2003. This calculation method comes from the idea of researchers from the University of California, Davis. They use PPPI ratios to estimate the variable costs of different research years. There is a reference absence here, I rechecked the manuscript and made some corrections. The omitted citation [18] and [19] of “reference research from UC Davis” is supplemented both on lines 165-167 and in the reference section.
Point 12: Line 163, is the PPPI specific to agricultural product?
Response 12: Thanks for your careful suggestions here. I really agree with you. Actually, Producer Price Payment Index (PPPI) is not specific to agricultural products here. It’s a common index for all commodities. Due to there is no PPPI data only for crops, I just use the common ones for estimation instead. If in the future I can find historical data for Crop PPPI or there are good Crop PPPI estimation models, I will try to correct my method in the future researches.
Point 13: Line 213, since you are considering a broad interpretation of costs as to include environmental costs you should not ignore the scarcity cost of water. See Kampas, A. and S. Rozakis (2017). "On the Scarcity Value of Irrigation Water: Juxtaposing Two Market Estimating Approaches." Water Resources Management 31(4): 1257-1269.
Response 13: Thank you for your good suggestions, professor. I downloaded the reference you provided and read through the whole paper. This paper juxtaposes two different market approaches that can be used for assessing the water scarcity rents, namely the supply and demand based approaches. It’s really an advanced model. But I only found the author provided the comprehensive model for the scarcity costs of irrigation water, but not for each crop. Thus,I tried my best to look for other researches which are concern with water scarcity rents, but still can’t hit the target. Due to the limitation of revising period, I may have to do the research of water scarcity rents for each crop in the future. Second, for the sake of preciseness, I pointed out this deficiency in the conclusion part.
Point 14: Lines 225- 249, Check the accuracy of the reference’s list.
Response 14: The reference list is rechecked and errors are corrected as shown in the revised manuscript.
Reviewer 4 Report
Overall this seems to be a well designed and well written paper. As such, it deserves serious consideration for publication. It provides a brief review of some of the major literature. But this is perhaps a bit too brief. It mentions the previous studies without actually specifying what each one’s contribution was, which makes it harder to understand what additional value this paper is offering.
The model is trying to get at the value of water for irrigation and so does not explicitly take into consideration the price of water. This can be compared to the value after the fact to see if in fact the value exceeds the cost. That is fine, but should be specified explicitly.
In terms of Equation 2, how is EP calculated or from where is it derived?
There are two equations labeled Equation 9. (lines 143 and 169). Obviously the second one should be labeled 10.
In Equation 9 (line 143), I’m still not sure why the denominator is twice the difference between high irrigated yields and general non-irrigated yields. Can the authors please offer an explanation for why this is the case?
In lines 166-167 the authors state that crop variable cost data was estimated by researchers from the University of California with “ good results.” The authors should provide a citation for this claim and should explain what “good results” actually means, as it is very subjective and vague.
In the conclusions, the authors claim that the model uses the actual profit of irrigation. But they also note that the model does not use fixed costs. So, the model is giving an economic return on water, but not necessarily a profit from water, no? Or am I not understanding something here?
Finally, there are a few minor spelling and grammatical errors. Nothing major. One specifically that could lead to confusion is in line 38. The authors write “… especially for acid areas lack of water resources.” I believe they meant something like “ especially for arid areas lacking water resources.”
Author Response
Response to Reviewer 4 Comments
Thank you very much for your review of the paper. According to your suggestions, I’ve made detailed revisions after careful consideration as follows.
Point 1: Overall this seems to be a well designed and well written paper. As such, it deserves serious consideration for publication. It provides a brief review of some of the major literature. But this is perhaps a bit too brief. It mentions the previous studies without actually specifying what each one’s contribution was, which makes it harder to understand what additional value this paper is offering.
Response 1: According to your comments, I made following revisions in order to enrich the literature review and reveal what additional value the EVIW model is offering.
First, some previous studies are supplemented to enrich the literature review. Changes are made as follows: ① In lines 25-31: several researches employed fundamental economic theories for economic valuation of water use. Young (1985) illuminated theories about economics of water resources such as the definitions of long-run and short-run value of water and their distinctions. The “equimarginal principle” involved in the maximization of efficiency and utility on water resource development and allocation were explained. Young also interpreted some theories about economic benefits related to location, quantity and time, and discussed the influences on benefits estimates made by factors like geographic location. ② In lines 36-40: Bate and Dubourg (1997) performed a net-back analysis on irrigation water demand in the East Anglia region of the U.K. The simple net-back model is introduced first, before existing data is used to estimate farmers' willingness to pay for irrigation water, both with and without the effects of agricultural and trade subsidies. This estimate is then compared with estimates of the long-run marginal cost of water. In lines 44-51: Many studies have conducted focusing on the economic valuation of water considering different uses. Frederick, et al (1996) divided the application of fresh water into four kinds of withdrawal and instream uses and estimated the value of water in these applications. In addition, the estimation of value on irrigation water was carried out using the data for 22 crops in different regions of the United States. Annes (2015) introduced the water distribution situation of Colorado River basin and discussed how to estimate the economic value of water in different regions of Colorado river basin. She estimated the value of agricultural, urban, and industrial water uses in different regions of the basin.
Second, each literature’s contribution is re-described. For example, in lines 40-42, the former description is changed to “Merrett (2002) provided a clear and succinct account of the linkage between the cost of irrigation water to farmers and the volume of water that they use in order to discuss how the use of water for farmers was impacted by irrigation water cost”.
Third, the remarks of previous studies are given in lines 56-64 so as to point out the shortcomings to be improved. The above literature elaborates several methods of estimating water economic value from different angles. The most direct is comparing recent market transaction prices for similar types of water transfers in the same area. However, this approach requires the availability of and collection of actual water transaction data in a specific region. Another approach used to assess the value of agricultural water is using water-crop production functions, based on the relationship between irrigation water and crop yield. However, Assumptions for this approach must be made about the level of other crop production inputs including fertilizers, pesticides, and labor. Furthermore, these methods are labor and data intensive and limited to locations and crops where accurate, up-to-date water-crop production functions are available.
Fourth, how this paper improves the research models is explained as follows : to avoid the above shortcomings, the EVIW model is constructed by improving the NRTW model put forward by Schuster, which is based on “Cost-benefit” Analysis and intends to find the net returns of agricultural water use brought to the crops. However, this method only deals with the whole economic value of agricultural water use, ignoring the differences between different kinds of crops. Besides, the overall cost is used for calculation without concrete decomposition. The EVIW model is a has taken a series of improvements to make the accounting of each income and cost item more accurate and practical. For example, as the corresponding cost items for different kinds of crops are quite different, it is needed to keep useful cost items and remove unnecessary ones. In other words, it’s an improved approach combining Cost-benefit Analysis with Cost Decomposition Method for assessing the economic value of irrigation water for different crops. The benefits of the EVIW model are: (1) it is accessible across broader scales and less expensive than the crop-water production function method; (2) it can be applied to the many regions where data is not available for the market price comparison method.
Point 2: The model is trying to get at the value of water for irrigation and so does not explicitly take into consideration the price of water. This can be compared to the value after the fact to see if in fact the value exceeds the cost. That is fine, but should be specified explicitly.
Response 2: According to your suggestion, the comparison between the economic value and water price is taken into consideration and specified explicitly in the “Results and Analysis” part.
Point 3: In terms of Equation 2, how is EP calculated or from where is it derived?
Response 3: Thanks so much for your careful findings here. I really agree with you. After I rechecked my research, I found EP is obtained by multiplying rainfall P with rainfall coefficient a. rainfall P can be obtained by yearly statistical data. For rainfall coefficient a, when P ≤ 5 mm, a = 0; when P = 6-50 mm, a = 0.8-1.0; when P > 50 mm, a = 0.70-0.80 [14]. This explanation is supplemented into the article.
Point 4: There are two equations labeled Equation 9. (lines 143 and 169). Obviously the second one should be labeled 10.
Response 4: Thanks for your careful check. The other equation is labeled 10.
Point 5: In Equation 9 (line 143), I’m still not sure why the denominator is twice the difference between high irrigated yields and general non-irrigated yields. Can the authors please offer an explanation for why this is the case?
Response 5: The Benefit Sharing Coefficient Method (BSCM) refers to the ratio of crop benefit increase caused by irrigation to total crop yield benefit increase. In Equation 9 (line 143), the numerator is “Y2-Y1” added by “Y4-Y3”, which means two times of yields difference caused by irrigation. So the denominator should be twice the difference between high irrigated yields and general non-irrigated yields. Besides, in order to make the equation easier to understand, I change the previous equation to .
Point 6: In lines 166-167 the authors state that crop variable cost data was estimated by researchers from the University of California with “ good results.” The authors should provide a citation for this claim and should explain what “good results” actually means, as it is very subjective and vague.
Response 6: For reference absence, I rechecked the manuscript and made some corrections afterwards. For example, the omitted citation [20] and [21] of “reference research from UC Davis” is supplemented both in lines 165-167 and in the reference section.
Point 7: In the conclusions, the authors claim that the model uses the actual profit of irrigation. But they also note that the model does not use fixed costs. So, the model is giving an economic return on water, but not necessarily a profit from water, no? Or am I not understanding something here?
Response 7: Thanks for your good suggestions. After rechecking, I changed “the actual profit of irrigation water” into “economic return on irrigarion water”.
Point 8: Finally, there are a few minor spelling and grammatical errors. Nothing major. One specifically that could lead to confusion is in line 38. The authors write “… especially for acid areas lack of water resources.”I believe they meant something like “especially for arid areas lacking water resources.”
Response 8: In regards to the language blemishes and grammar errors, I asked an English-proficient professor to help me go through the article and mark the wrong parts, and then I corrected all the errors. The modification of language and grammar is highlighted in the revised draft. If necessary, I will ask expert editors from MPDI for language editing.
Round 2
Reviewer 1 Report
You are right that your manuscript is about optimal allocation of scarce water (and in the title you should use optimal instead of sustainable), but your optimization is arranged within only the framework of standard economics. It means that, it is incomplete as it comes from only incremental economic benefits of different products caused by changes in irrigated water.
I have nothing against your article, but you should offer it to some economic journal and not to the transdisciplinary Sustainability journal.
Author Response
Response to Reviewer 1 Comments
Point 1: You are right that your manuscript is about optimal allocation of scarce water (and in the title you should use optimal instead of sustainable), but your optimization is arranged within only the framework of standard economics. It means that, it is incomplete as it comes from only incremental economic benefits of different products caused by changes in irrigated water. I have nothing against your article, but you should offer it to some economic journal and not to the transdisciplinary Sustainability journal.
Response 1: Thank you very much for your recognition of my paper. After in-depth consideration of your comments, I think this paper still has two main problems to solve: ① there is no cut-in point with the theme of "sustainability"; ② the topic of this paper is still ambiguous. To solve the above two issues, I’ve made detailed revisions as follows.
First, interpreting the research significance. I’ve made the following modifications based on the original manuscript:
I. The research background and role of crop economic value estimation plays in “water-crop” sustainable development is added between lines 22-31: With the rapid development of economy and society and the continuous growth of population, the irrigation water demand of arid areas is also increasing. The contradiction between the increasing demand for water resources and the inadequate availability of water resources in arid areas has become more and more obvious, which largely restricts the sustainable development of "economy-society-ecology" in the region. As the agricultural economy in arid areas is severely constrained by water shortage, the maximization of irrigation water value is of great significance to agricultural development in arid areas. For example, to achieve the goal of sustainable utilization, the economic value of water can be used as a starting point for the optimal allocation of crop irrigation water and efficient planning of crop planting area, which can provide a basis for improving the utilization efficiency of irrigation water.
II. The research objective is elaborated between lines 34-38: This paper aims to create an effective model to reasonably estimate the economic value of irrigation water by means of quantitative analysis, which provides a research basis for the economic value maximization of water resources, efficient use of irrigation water and crop planting planning. Furthermore, the model created in this paper provides an effective technical basis for "water-crop" sustainable development and regional economic stability in arid areas.
III. In conclusion part, the influence to "water-crop" sustainable development brought by the EVIW model is explained in lines 260-267: The EVIW model makes contributions to the following three aspects to achieve "water-crop" sustainable development and regional economic stability in arid areas: First, by comparing the economic value of irrigation water of different crops, farmers can choose to plant crops with greater economic value of water to overcome the problem of water shortage and realize the sustainable utilization of farmland. Second, combined with plant planning, the estimated economic value of irrigation water can be used as a judgement criterion for the optimal allocation of crop irrigation water. Third, the estimated economic value of irrigation water can provide a basis for improving the utilization efficiency of irrigation water.
Second, in the last response to your comments, I explained wrongly that this paper is a topic of optimal allocation of irrigation water. I have to say sorry for this arbitrary viewpoint. Because actually this paper is trying to construct a model applicable to different crops to evaluate the irrigation water economic values but not to optimize the allocation of irrigation water. This is a more direct approach and easier way to evaluate the economic value of crop irrigation water compared with other complex and huge models. I hope to verify the accuracy and feasibility of this method to compare this method with the acknowledged approaches. If it’s really feasible to use for evaluation of water economic values, we can use it for further exploration.
Besides, initially I prefer to build a crop irrigation water allocation model based on the economic value of irrigation water. However, the applicability range of the two models is inconsistent. This paper is trying to construct an “universal” economic value estimation model (EVIW model) for assessing the economic value of water used for crop irrigation, especially in acid areas lacking water resources, while the optimal allocation of water has to consider the facts of the specified study area. As a result, temporarily I could only make a model research for a specified area instead of all kinds of areas. If the EVIW model constructed in this paper is valuable, I would do a future research about combining the EVIW model with a new irrigation allocation model. For the question of topic ambiguity, “the sustainable utilization of” is removed in order to avoid unnecessary misunderstanding.
Criticism and suggestions are welcome. Hope to hear from you again, dear professor. Thank you again for your review.
Reviewer 3 Report
no further comments
Author Response
Thank you very much for accepting my paper, dear professor.
Round 3
Reviewer 1 Report
Dear authors, I am sorry to repeat my former conclusions. Your manuscript might be interesting for some economic journals, but has only negligible contribution for the transdisciplinary Sustainability journal. By your model, targeted on maximization of irrigation water value, you are only emulating market allocation. And the market is socially and environmentally blind, as is your manuscript. You are not addressing related problems of social inequities and environmental externalities caused by applying your modelling results. So, please try to offer your article to some economic journal where I am sure you may be successful. At the same time, try to study social and environmental connotations of our globalized market economic world.